# Pyrene-Based AIE Active Materials for Bioimaging and Theranostics Applications

**DOI:** 10.3390/bios12070550

**Published:** 2022-07-21

**Authors:** Muthaiah Shellaiah, Kien-Wen Sun

**Affiliations:** Department of Applied Chemistry, National Yang Ming Chiao Tung University, Hsinchu 300, Taiwan; muthaiah1981@nycu.edu.tw

**Keywords:** aggregation-induced emission, pyrene, cellular imaging, small molecules, polymers, theranostics, sensors

## Abstract

Aggregation-induced emission (AIE) is a unique research topic and property that can lead to a wide range of applications, including cellular imaging, theranostics, analyte quantitation and the specific detection of biologically important species. Towards the development of the AIE-active materials, many aromatic moieties composed of tetraphenylethylene, anthracene, pyrene, etc., have been developed. Among these aromatic moieties, pyrene is an aromatic hydrocarbon with a polycyclic flat structure containing four fused benzene rings to provide an unusual electron delocalization feature that is important in the AIE property. Numerous pyrene-based AIE-active materials have been reported with the AIE property towards sensing, imaging and theranostics applications. Most importantly, these AIE-active pyrene moieties exist as small molecules, Schiff bases, polymers, supramolecules, metal-organic frameworks, etc. This comprehensive review outlines utilizations of AIE-active pyrene-based materials on the imaging and theranostics studies. Moreover, the design and synthesis of these pyrene-based molecules are delivered with discussions on their future scopes.

## 1. Introduction

Aggregation-induced emission (AIE) is a unique phenomenon in organic luminophores, fluorescent dye molecules, nanoparticles, carbon dots, organic-inorganic nanocomposites, etc. [1,2,3,4,5,6]. In general, AIEgens are found to be non-emissive or weakly luminescent materials in their solution state and exhibit enhanced emission in the aggregated stage via twisted intramolecular charge transfer (TICT) or restricted intramolecular rotation/motion (RIR/RIM) mechanisms [7,8,9,10]. 

The AIE-active materials have been applied in ground-breaking applications, such as chemo/biosensors, hazardous detection, optoelectronic devices, therapeutics, theranostics and stimuli responsive imaging studies. [11,12,13,14,15,16,17,18,19,20]. In particular, applications of AIE-active materials in opto-electronics, biosensors, imaging, therapeutic and theranostics are not only in great demand but also make a social impact [21,22,23,24,25,26]. For example, tetraphenylethylene (TPE) derivatives were noted as unique AIE-active materials towards sensing, bioimaging, therapeutics and theranostics utilities [27,28,29,30]. 

Therefore, researchers have developed various organic fluorophores consisting of polycyclic/branched aromatic hydrocarbons [31,32,33,34]. Among these polycyclic aromatic hydrocarbons, pyrene is noted as a highly emissive organic luminophore having a flat aromatic scaffold with four fused benzene rings [35,36]. Due to the *π–π* stacking property, the monomer and excimer emission of pyrene-based derivatives displays distinct emissive properties in the presence of analytes, thereby, acting as stimuli responsive materials [37,38,39,40,41,42]. Likewise, pyrene containing metal-organic frameworks (MOFs) and metallocages are highly emissive materials due to their self-aggregation nature, which were utilized as emitters and reactive oxygen species (ROS) generators [43,44,45,46,47]. 

Synthetic modifications of the pyrene scaffold lead to the formation of weakly and/or highly emissive derivatives that have been utilized in various studies. In fact, the weakly emissive pyrene-based derivatives may possess TICT or photo-induced electron transfer (PET), which inhibit the aggregation of those molecules [48,49]. Interactions among analytes or stimuli towards those weakly emissive pyrene derivatives inhibit the TICT/PET processes, which in turn allows the aggregation of molecules and leads to emission enhancement. 

This enhanced emission is noted as aggregation-induced emission enhancement (AIEE) [50,51]. For instance, interactions between water and weakly emissive molecules led to the formation of a highly emissive aggregation state via H-bonding [52]. During AIEE, the pyrene derivatives underwent excimer formation, which led to structures of feasible *H*- or *J*-type aggregate [53,54,55,56,57,58]. By definition, molecules stacked in a predominantly face-to-face arrangement is called *H*-aggregation. On the other hand, molecular stacking that displays in a head-to-tail arrangement is known as *J*-aggregation [57,58] as seen in Figure 1.

AIE of pyrene derivatives have found various applications, such as sensors, opto-electronic devices, imaging and theranostics studies [58,59,60,61]. Numerous pyrene derivatives can deliver both sensing and AIE properties simultaneously [49,58]. Likewise, a few sensory reports on pyrene-based derivatives did not address the AIE of the molecules [58]. Upon the conjugation of pyrene with dye molecules and peptides, the feasible in vivo/in vitro aggregation of those molecules can be applied in bioimaging and theranostic studies [62,63,64]. 

For example, Wang and co-workers reported the conjugation of bis-pyrene derivatives with dyes and peptides for bioimaging and therapeutics [64]. Similarly, the AIE of pyrene-based small molecules can be utilized for in vivo/in vitro imaging applications [35]. There are already many reviews on the aggregation-induced emission of several reported molecules in optoelectronic, bioimaging and theranostics [1,2,3,4,5,6,7,8,9,10,11,12,13,14,15,16,17,18,19,20]. However, reviews on the pyrene-based AIE-active materials for bioimaging and theranostics applications are not readily available and will be delivered in this article (Note: considering the journal scopes, other optoelectronic applications of the AIE-active pyrene-based materials are omitted in this review).

A representative scheme of using the AIE-active pyrene-based materials towards bioimaging and theranostics studies and representations of feasible *H*- or *J*-aggregates are displayed in Figure 1. In this review, the advantages and limitations of the AIE-active pyrene-based molecules prompted the future research directions. Moreover, the design and synthesis of those AIE-active pyrene derivatives are delivered, and their future scopes are discussed.

## 2. Bis-Pyrene Derivatives for Bioimaging and Theranostics

In this section, the effectiveness of AIE-active bis-pyrene (BP) derivatives towards bioimaging and therapeutics are discussed with reported evidence. Applications in enhanced imaging and theranostics were successfully demonstrated by prof. Hao Wang’s research group [65,66,67,68,69,70,71,72,73,74,75]. Bis-pyrene derivatives **BP1**–**BP4** were produced via design of bis-pyrene derivative with 1,3-dicarbonyl, pyridine-2,6-dicarbonyl, oxaloyl and benzene-1,4-dicarbonyl linkers [65]. In the solution state, the **BP1**–**BP4** displayed weak emission with low quantum yields (Φ < 1.7%). By adding water, the solid state AIE of **BP1** and **BP2** was enhanced 30-fold with a Φ value reaching 32.6%. 

On the contrary, the molecules **BP3** and **BP4** displayed poor AIE enhancement with a Φ value of approximately 3.1%. The observed distinct solid state AIE changes suggest that the bis-pyrene scaffolds are due to the formation of diverse aggregation frameworks. In solid state, the **BP1** and **BP2** form the *J*-type aggregates, whereas the **BP3** and **BP4** tend to form the *H*-type aggregates instead. Figure 2A displays the excimer formation by the **BP1** and **BP2** to afford the *J*-aggregates. Transmission Electron Microscopy (TEM) studies revealed dot shaped nano-aggregation by the **BP1** and **BP2** with sizes of 2–6 nm in diameter. 

On the other hand, the **BP3** and **BP4** aggregates displayed sheet-like morphology with dimensions of 5–10 nm in width and 20–100 nm in length. To stabilize the nanoaggregates of the **BP1** and **BP2**, surfactant F108 was utilized and displayed the long-term imaging ability. Both the **BP1** and **BP2** nanoaggregates exhibited high photostability and pH stability between pH values of 5–10. To investigate the low toxicity of **BP1** and **BP2** nanoaggregates, cellular imaging studies were conducted in KB cells (human oral epidermoid carcinoma).

The LysoTracker red-labelled fluorescence images reveal that the **BP1** and **BP2** nanoaggregates are mainly located on the lysosomes after 3-h incubation, as seen in Figure 2B. Thus, it is concluded that both the **BP1** and **BP2** nanoaggregates act as the lysosome targeted nanoprobes. This is an impressive work regarding the lysosome imaging and tracking by judging from the design and obtained results. A pH-responsive bis-pyrene-conjugated polymer **C1** (represented as “**P-BP**” in the original report; see Figure 3a) was synthesized via hydrophobic interactions and self-assembled in a nanoparticles shape [66]. The formation of nanoparticles and pH-responsive fluorescent enhancement are attributed to the AIE and *J*-aggregate nature of the bis-pyrene moiety as detailed next.

The hydrodynamic diameter of **C1** determined via the dynamic light scattering (DLS) technique increased from 41.7 to 138.2 nm with decreasing pH values from 7.4 to 5. The emission peak resulted from the polymer nanoparticles formed *J*-type aggregates was blue-shifted from 527 to 418 nm as the pH values changed from pH 7.4 to 5. This was attributed to the separated bis-pyrene from the polymer system via ionization. Thereby, it behaves as a pH-responsive probe and can be applied in bioimaging. 

To achieve pH responses with an extended shift in fluorescence, the polymer **C1** was encapsulated with the Nile red (NR) dye towards the construction of fluorescence resonance energy transfer (FRET) conjugate **C1/**NR (Donor/Acceptor). After encapsulating the NR dye (emission peak at 628 nm) over the polymer, the nanoparticles displayed red emission by means of the FRET effect. Between pH values of 7.4 to 5, the emission peak of **C1/**NR conjugate at 635 nm was gradually quenched, which was accompanied by an increasing emission at 418 nm from the bis-pyrene in solution state. 

To explore the endocytic microenvironmental pH response, the low toxic **C1/**NR conjugate was incubated in human primary glioblastoma (U87) cells and investigated by confocal laser scanning microscopy (CLSM). Time-dependent cellular imaging studies between 10–90 min displayed the colocalization of bis-pyrene and lysosomes (labelled with LysoTracker Green DND-26) via blue emissive cell lines, which confirmed the microenvironmental pH sensing ability of the **C1/**NR in living cells. 

This probe can be regarded as a unique and potential candidate for pH monitoring in the endocytosis process. Additionally, two bis-pyrene monomers, **C2** and **C3** (see Figure 3a; noted as the “**BP1** and **BP2**” in the original report**)** were dispersed in hydrophobic pH-sensitive mPEG-g-poly(aminoesters) graft copolymers (PbAE) micelles and engaged in lysosome imaging at pH < 6.0 [67]. Due to polymer trapping of the bis-pyrene derivatives, the BPs in PbAE did not produce any emission at 512 and 532 nm at pH 7.4. As pH values decreased to 6 or 5.5, the ionized polymer released the **C2** and **C3** and resulted in enhanced fluorescence at 514 and 533 nm, respectively, via the stimuli responsive AIE effect and *J*-aggregate formation. Bioimaging studies in Hela cells (the first immortal human cells ever grown in culture treated cancer cells) indicated that the above polymer dispersed bis-pyrene-conjugated system was effective in the pH-sensitive endocytosis process and in lysosome imaging.

Thereafter, bis-pyrene-conjugated cyanine dye **C4** (see Figure 3c; noted as “**BP-Cy**” in the original report) was self-assembled into vesicular nanoaggregates (size = 59.9 ± 9.8 nm as determined by scanning electron microscopy (SEM)) and utilized in the photo-acoustic (PA (definition: upon the irradiation of laser on the materials, it generates ultrasonic wave and delivers the images of distributed light energy absorption in the tissue)) imaging studies [68]. Due to conjugation of the BP with the cyanine dye and AIE effect, the **C4** exhibited an emission peak at 790 nm with intensity comparable to the indocyanine green (ICG; Food and Drug Administration (FDA) approved PA agent) and displayed a PA intensity enhancement by a factor of 1.6-times higher. 

During laser irradiation (power density of 490 mW at 790 nm), heating–cooling curves were recorded, which showed a 9.2-times higher heat conversion efficiency (*η*) than that of ICG. The proposed **C4** nanoaggregates were engaged as the long-term cell-tracking PA contrast agents in MCF-7 cells (human breast cancer cells) and were effective up to the seventh-generation incubation (for each generation 36 h incubation was considered). The first treated cell lines were regarded as the first generation and used in the PA imaging. The other 50% of cells in a fresh growth medium were transferred and incubated for 36 h and were noted as the second generation. 

This process is continued up to the seventh generation. From the in vitro PA imaging studies, the half-life of **C4** nanovesicles was established as 5 days, which was much higher than the ICG (0.5 day), thereby, confirming the long-term PA imaging ability. To verify the in vivo PA imaging ability, the breast-tumor-xenografted mice were treated with the **C4** and ICG. The **C4** displayed PA signals in the tumor region after 2 h accompanied with enhanced intensity up to 12 h. The signal was then decreased but still detectable until 72 h. On the other hand, PA signals from the ICG-treated tumor xenografted mice reached maximum after 2 h and were only detectable up to 4 h. This work on the bis-pyrene-cyanine conjugate-based nanovesicles for long-term PA imaging can be regarded as an impressive one and can be applied in future biomedical devices.

A FRET-based co-assembled nanosystems was reported using the bis-pyrene **C5** as donors and the near infra-red (NIR) emissive dyes as acceptors (see Figure 4a; noted as “**BP**” in the original report) to apply in the in vivo two photon fluorescence imaging (TPFI) [69]. The weak fluorescent **C5** was initially self-assembled to provide turn-on green fluorescence because of the *J*-type nanoaggregate formation (with a size of 35.8 ± 2.8 nm) from the bis-pyrenes and AIE effect. 

It was further co-assembled with the acceptor NIR dyes (NR = Nile red; DCM = 4-(dicyanomethylene)-2-methyl-6-(4-dimethylaminostyryl)-4H-pyran; RhB = Rhodamine B and Rh6G = Rhodamine G) to deliver FRET-based spherical nanosystems (with a size of 34.1 ± 1.8 nm), which was utilized in the TPFI studies. Among these **C5**/NIR-dyes composited nanosystems, the **C5**/NR is more effective in the TPFI interrogations. During the development of low toxic **C5**/NR nanosystems, it was found that the PL peak at 510 nm was quenched with increasing emission at 630 nm as the NR dye (0–160 µM) was gradually added into 40 µM nanoaggregates. Upon incubation of the **C5**/NR nanosystems in Hela cells, the green channel and red channel emissions were recorded at 495–540 and 575–630 nm, correspondingly. 

From the mock-tissue investigations, the high penetration ability of the C**5**/NR nanosystems were observed up to 2200 µm in depth, which can be a potential candidate for TPFI of mice cartilage (a non-vascular type of supporting connective tissue that is found throughout the body). This design can be regarded as an innovation based on the AIE and FRET mechanism. Thereafter, NIR-light-propulsive Janus-like nanohybrids were constructed using the bis-pyrene nanoaggregates and engaged in the photothermal tumor therapy (PTT) [70]. In this report, the Au stars and BP molecules were bound with the mercapto propyl triethoxysilane (MPTES) and PEGlated with mPEG_5000_-Mal (1 mL, 0.1 wt%) to form the Au-BP@SP Janus nanohybrid systems. 

By varying the MPTES (20, 30 and 40 µL) and BP (100 and 200 mg) concentrations, six different nanohybrids (noted as the Au-BP1-6@SP) were developed, which demonstrated enhanced emission at 520 nm with increasing BP concentrations. The SP layer thickness was found to increase from 9 to 28 nm in the Au-BP4@SP and Au-BP6@SP. Thereby, the Janus nanohybrids formed by the Au stars and BP aggregates, and TEM studies confirmed that the size of nanohybrids can be increased by increasing the BP and MPTES concentrations. 

From the above observations, the nanohybrid Au-BP7@SP with large amount of BP (800 mg) and MPTES (30 µL) were synthesized to monitor active cellular motion in the microscopic thermophoresis investigation (thermophoresis is a force generated by the temperature gradient between the hot gas and the cold wall affecting the particulate movement towards the cold wall). The diameters of Au-BP7@SP and BP aggregates in nanohybrids were established as 227 and 157 nm, respectively. Upon irradiating the Au-BP7@SP nanohybrid with a NIR laser (λ_ex_ = 810 nm), the nanohybrid showed a controlled active motion and produced diverse temperatures (between 38.0–55.7 °C) in the MCF-7 tumor-bearing mice to kill the cancerous cells via PTT, which was visualized using two-photon fluorescence microscopy (TPFM). 

In order to record the TPFM images, concentration-dependent aggregation-induced emission (CAIE) of the BP molecules played a vital role in PTT. Thus, this work can be regarded as an innovation towards PTT based on the BP-aggregate-facilitated nanohybrid system. A bis-pyrene and peptide containing supramolecular module **C6** (noted as “**BKR**” in the original report) was reported for the Ca^2+^-regulated self-assembly of peptides over cell surfaces, which can kill cancer cells effectively [71]. In a water-hexafluoroisopropanol (H_2_O/HFIP) media, the **C6** nanoparticles, with enhanced and red-shifted fluorescence from 519 to 528 nm, were formed with a size of 30.2 ± 5.3 nm due to self-aggregation. 

The incubation of low toxic **C6** nanoparticles with Ca^2+^ ions led to the formation of nanofibers with an average diameter of 10.2 ± 3.4 nm, which can be further applied in cell surface-based in situ transformation. In the presence of Ca^2+^, incubation of the **C6** nanoparticles in U87 cell lines (brain tumor cells) resulted in better adhesion of the **C6** to cancer cell surfaces via nanofiber formation and caused the death of cells. This an impressive work towards cancer theranostics; however, details regarding the AIE effect on imaging studies require further investigation.

Subsequently, the bis-pyrene constituted polymer-peptides (**PBP_1–3_**) were engaged in the thermo-responsive proliferative suppression of SK-BR-3 cell lines (breast cancer cells) [72]. The functional pairing motif HBP was covalently linked to the temperature-sensitive polymer PNIPAAm (poly(N-isopropylacrylamide)), which collapsed at 40 °C to stop the aggregation. After cooling to 35 °C (lower critical solution temperature (LCST)), the polymer entered into an extended state and released the HBP in aqueous media to enhance the AIE-based emission (from *J*-type nanoaggregates of BP and *π–π* stacking) for monitoring human epidermal growth factor receptor-2 (HER2) from the breast cancer cells inhibition. Figure 5 shows the structures of thermo-responsive polymers (the iodine polymer was used to distinguish samples from the other membrane components) and their corresponding SK-BR-3 cellular images. Based on the above results, this work is impressive in PTT.

Figure 6A displays detection of the intracellular glutathione (GSH) distributed in tumor cells, such as MCF-7, by the bio-orthogonal reaction of the bis-pyrene-lined cyanine moiety [73]. The cyanine (Cy) and bis(pyrene) (BP) motifs are covalently linked through a disulfide linkage. The linkage is affected during the bio-orthogonal reaction to release the cyanine dye and BP-nanoaggregates, which results in emission at 520 nm with a red background at 820 nm. In fact, the GSH is attached on the cyanine dye (see Figure 6A) as evident from the MALDI-TOF mass data (see Figure 6B). 

The AIE effect from the BP-nanoaggregates exhibited strong turn-on green emission at 520 nm with a 36-fold enhancement. Due to the bio-orthogonal reaction, the GSH in the tumor MCF-7 cells are tracked in both green and red channels as seen in Figure 6B. This work was also demonstrated by the in vitro mice MCF-7 cells investigation, thereby, setting a milestone in the tumor diagnostics studies. However, more attention focused on the importance of the BP-nanoaggregate-tuned emission enhancement is required.

Subsequently, dendritic peptides conjugated BP moiety (noted as the **DPBP**) was used in temporary in situ intracellular monitoring of autophagy process (natural degradation of the cell that eliminates unnecessary components via a lysosome-regulated mechanism, which suppress cancer) [74]. As shown in Figure 7A, the intracellular autophagy-related cysteine proteases (ATG4B) was monitored through enzymatic cleavage of peptides, which released the BP to emit turn-on fluorescence from the AIE effect. The specificity of low toxic **DPBP** towards the ATG4B was demonstrated in living cells and zebrafish investigations. They engaged the MCF-7, Rapa (rapid adenovirus production and amplification), siRNA (double stranded RNA) cell lines and LC3-autophagy marker for the ATG4B detection as depicted in Figure 7B. Based on the AIE-based turn-on in vivo/in vitro autophagy detection and cancer cell tracking ability, the **DPBP** can be attested as a potential candidate for cancer diagnostics.

By means of an intraparticle FRET mechanism, the bis-pyrene doped cationic dipeptide nanoparticles (**BP-CDPNP-RB**) was proposed for the two-photon activated photodynamic therapy (TPA-PDT) [75]. Herein, the cationic dipeptide (H-Phe-Phe-NH_2_·HCl, CDP) was doped with the BP, which acted as an energy donor as well as the two-photo fluorescence dye. The rose bengal (RB) served as an acceptor (see Figure 8). Due to the FRET mechanism, irradiation with two photon laser (λ_ex_ = 810 nm) induced energy transfer from the BP to the NIR dye RB to enhance generation of singlet oxygen in cell culture and inhibited cancer growth as demonstrated in the MCF-7 imaging studies. Although this report appears to be impressive, discussions regarding the AIE effect of BP are missing. The optical properties and imaging applications of the important bis-pyrene compounds/systems discussed in this section are summarized in Table 1. 

Other than the aforementioned bis-pyrene-based theranostics research in this section, some of the BP-derivatives were also reported for fluoride detection and multi-responsive fluorescent switches [76,77], which attested the BP as a potential candidate towards optoelectronic and biological applications.

## 3. AIE Active Pyrene Conjugates for Bioimaging

Jana and co-workers reported the 1-(hydroxyacetyl)pyrene as a phototrigger of alcohols and phenols [78]. In this, the pyrene caged adenosine (**C9**; see Figure 9a; numbered as “**9**” in the original report) was photo-triggered to release the **C7** and **C8** (see Figure 9a; numbered as “**1** and **3**” in the original report) and to engage in in vitro bioimaging studies. The imaging results may have resulted from the mild aggregation of released pyrene scaffold in the cellular system. However, no evident AIE effect were found. Thereby, it cannot be considered as an AIE-based imaging candidate at the current stage, and this requires further investigations.

Ghosh et al. reported a pyrene derivative **C10** (see Figure 9b; noted as “**A3”** in the original draft) lysine-triggered fluorescence enhancement by means of the AIE effect in the MCF-7 cellular imaging studies [79]. During the Lys-triggered AIE, the particle size increased from 24.7 to 214 nm through conversion from the dynamic excimer to static excimer and acted as an exceptional in vitro imaging agent. By means of this AIE-based approach, the detection limit (LOD) of Lys was calculated as 3 nM (nM = nanomolar). After incubation of the **C10** with the Lys in MCF-7 for 2 h, a strong blue emissive cell line was visualized because of the Lys-triggered AIE effect. This is an impressive report for Lys quantitation based on the AIE effect; however, more investigations towards therapeutics are still needed. 

Subsequently, a pyrene-fluorescein conjugate **C11** (see Figure 9c; noted as “**FHPY**” in the original report) was described for AIE-tuned in vitro imaging applications [80]. In the AIE studies (*f_w_* = 0–90%), the *π–π* stacking of probe **C11** led to formation of *J*-type aggregations and strong fluorescence at 470 nm (for *f_w_* = 70%; quantum yield (Φ_F_) = 97%). Incubation of the **C11** in Hela cells for 24 h displayed a blue emissive cell line, which suggested its biocompatibility. The AIE-based in vitro imaging of C11 attests its feasible bioimaging applicability in the future.

Lalitha and co-workers constructed the organogels by means of self-aggregation of the pyrene coupled coumarin derivatives **C12a**–**c** (see Figure 9d; noted as “**5a**–**c**” in the original report) [81]. The organogels were formed because of self-aggregation and *π–π* stacking induced nanofibers formation. Note that these pyrene-based derivatives **C12a**–**c** were engaged in fibroblast L929 cellular imaging. However, this lacks clear evidence of the AIE effect in the bioimaging studies. Likewise, Sun et al. reported a pyrene-lactose conjugate **C13** (see Figure 9e; noted as “**Py-Lac**” in the original report) for two-photon imaging studies via self-assembly and *π–π* stacking [82]. 

The conjugate **C13** showed higher binding ability to the peanut agglutinin (PNA) lectins. Upon incubation in the Hep G2 cells (immortal cancer cell lines), green fluorescence is found to be predominantly located in cytoplasm as seen in Figure 10. This report discussed only the self-assembly but not the AIE. Thus, it cannot be considered as the AIE-based bioimaging. In contrast, Nie et al. reported two AIEgens, namely the **C14** and **C15** (see Figure 9f; note as “**Py-CN-N** and **Py-CN-S**” in the original report), for dual channel imaging and ratiometric sensing in living cells [83]. 

The AIE interrogations on the **C14** and **C15** revealed that both probes exhibited two emission bands at 460 and 570 nm by means of “monomer–dimer–multimer” formation, corresponding to “weak-blue-red” emission. The AIE property was well authorized from the DLS, time resolve photoluminescence (TRPL) and pH investigations. Incubation of the **C15** in Hela and Raw 264.7 (cancerous monocyte/macrophage-like cells) cell lines demonstrated the red channel imagining ability. On the other hand, incubation of the **C14** in living cells revealed the blue and red channel imaging utility and justified its ratiometric sensing applicability in living cells. This report can be regarded as a good work among the dual-channel bioimaging applications.

Panigrahi et al. reported a pyrene-based AIEgen **C16** (see Figure 11a; noted as “**TGP**” in the original report) that can form the cationic nanoaggregates for wash-free bacterial imaging and anti-microbial applications [84]. The low toxic cationic AIEgen having the C_3_-symmetry was synthesized by one-pot Schiff base condensation to form nanoaggregates in a water−DMSO mixture (*f_w_* = 80%). Photoluminescence, TRPL, SEM, DLS and zeta potential studies demonstrated the AIE and concentration-dependent positively-charged nanoaggregation of the **C16**. The positively-charged nanoaggregates interacted strongly with the bacterial surface (*E. coli K12* and *B. subtilis b* were used). 

The antibacterial activity illustrated in Figure 12A shows that an efficient electron transfer from the nanoaggregates to bacterial membrane takes place to generate the intracellular ROS to disintegrate the bacteria membrane. This work elaborated a pyrene-based Schiff base AIEgen towards bacterial imaging applications.

Dong et al. demonstrated the luminescent pyrene modified tetraphenylethylene probes **C17** and **C18** (see Figure 11b; noted as “**Probe 1** and **Probe 2**” in the original report) towards AIE-based detection of the cysteine (Cys) and homo-cysteine (Hcy) from the glutathione (GSH) and other amino acids [85]. Both probes displayed 2000-fold of fluorescence enhancement to the Cys and Hcy and were effective in a wide pH range from 3 to 10. 

The initial AIE of the probes in tetrahydrofuran (THF)-water was further enhanced in the presence of the Cys and Hcy at *f_w_* = 99% (water/THF; *v*/*v*) with the maleimide acting as a recognition group in both probes. As visualized in Figure 12B, incubation of the **C17** and **C18** probes in H1299 cells (human non-small cell lung carcinoma cell line) displays distinct blue emissive cells due to interactions with the intracellular thiols. Upon pretreated with the N-methylmaleimide, the intracellular thiols were removed, and therefore no fluorescence is observed. 

This bioimaging result suggests that these probes are effective for real time imaging of the thiols in living cells. However, information regarding the nanoaggregation features of the probes still needs to be clarified.

Recently, Kundu and co-workers synthesized a pyrene-based Schiff base **C19** (see Figure 11c; noted as “**ABzPy**” in the original manuscript) by reacting the 3,4-Diaminobenzophenone with 1-pyrenecarboxaldehyde in ethanol and utilized in AIE-based in vivo imaging studies [86]. The probe was in a form of concentration-dependent nanoaggregates in DMSO and exhibited strong emission at 575 nm (monomer emission at 450 nm). This might be due to existence of the *π–π* in the excimer state and inhibition of photo induced energy transfer (PET) present in monomer. The probe, **C19** nanoaggregates, was in a spherical shape with a size of 20−30 nm at a water fraction (*f_w_*) = 99% (water/DMSO; v/v; DMSO-dimethylsulfoxide) and exhibited excimer emission bands at 550 and 600 nm. 

By incubating the probe in Hela cells, together with the lysotracker red and lipid droplet tracker dye Nile red, its capability of targeting the lysosomes and lipid droplets was attested. As shown in Figure 13, the lysotracker red and supernatant of aqueous dispersion of the **C19** were witnessed in the lysosomes. Due to its greater hydrophobicity, nanoaggregates of the probe can localize the specific lipid droplets. Based on the above results, this small molecular pyrene scaffold can be regarded an innovation to be engaged in multiple bioimaging studies.

Pang’s research group proposed to use the pyrene–benzothiazolium and the pyrene-pyridinium dye molecules **C20**–**C24** (see Figure 11d,e) noted as “**1**, **2** and **1**–**3**” in the original reports) in the lysosome/cell-nucleus imaging and long-term tracking [87,88,89]. These dye molecules acted as effective markers in the tumor cell models and endocytosis process, which were interpreted in terms of the hydrophobic interactions instead of the AIE effect. 

In fact, due to the existence of the hydrophobic effect, these dye molecules might have formed aggregation to induce enhanced emission, which was not mentioned in the reports. Thereby, details of the bioimaging utility of the **C20**–**C24** are not discussed here. However, AIE effect of the **C20**–**C24** in bioimaging must be investigated in the future. Two Y-shaped pyrene containing DNA (deoxyribonucleic acid) nanoprobes (see Figure 14) were developed for intracellular detection of the microRNA (RNA = ribonucleic acid; miRNA plays vital role in gene expression via binding with messenger RNA (mRNA) in the cell cytoplasm) via red-shifted AIE from the pyrene monomer-excimer switch [90].

The **C25** and **C26** (are assigned as **Y-1** and **Y-2** in the original manuscript) are able to detect the miR-21 and miR-let 7a (two cancer promoting microRNAs), respectively, via observation of a red-shifted monomer emission band from 400 to 480 nm due to the excimer formation. From the PL-based studies, LODs of the miRNA by the **C25** and **C26** were established in picomolar (pM) levels (200 and 230 pM, respectively, with a linear range of 0–100 nM). This suggests feasible AIE-tuned detection of the intracellular miRNA. Incubation of the **C25** in the MCF-7 and Hela cells (a human cervical cancer cell line with miR-21 under-expression) demonstrated intracellular detection of the miRNA. Similarly, upon incubating the **C26** in the MCF-7 and A549 cell lines, selective intracellular recognition of the miR-let 7a can be visualized. 

Due to the DNA conjugation, both Y-shaped probes showed low toxicity and higher cellular uptake compared with those of ss-DNA and ds-DNA (ss-single stranded and ds-double stranded). These results suggest that this AIE-based design can be utilized as an effective strategy for detecting the cancerous miRNAs. Thereafter, Lee and co-workers reported the pyrene modified guanine cluster probes, which formed the DNA/RNA hybrid three-way junctions and applied in detection of the intracellular miRNA [91]. 

In the presence of RNA, the original blue emission at 455 nm from the probes was quenched accompanied by a red-shifted and enhanced yellow emission at 580 nm. It was attributed to *π–π* stacking and self-assembly/self-aggregation of the pyrene-based probes. Incubation of the hybrid probes in Hela cells displayed yellow emissive cell lines. Thereby, the probes are quite effective in bioimaging of the intracellular miRNA (miR-191; family of microRNA precursors found in mammals). The focus of this paper was majorly on the self-assembly rather than the AIE effect.

Li et al. demonstrated self-assembly of the amphiphilic 4-(4-(pyren-1-yl)butyramido)phenylboronic acid (**Py-PBA**) in a nanorod (NR) shape in DMSO-water mixture and applied in intracellular two-photon imaging of the sialic acid (SA; plays vital role in many biological and pathological processes) [92]. The monomer emission of the **Py-PBA** was at 400 nm. During the NR formation via self-assembly/self-aggregation, the excimer emission was red-shifted to 475 nm due to strong *π–π* stacking interactions. Two-photon imaging studies were conducted by incubating the **Py-PBA** NRs in the MCF-7 cell lines to perform in situ imaging and evaluation of the SA on the living cell surface. Moreover, the **Py-PBA** NRs also generate O_2_ under two-photon irradiation. Therefore, they can also serve as the photodynamic therapeutic agent. However, the results are interpreted based on *π–π* stacking and hydrophobic effect, thereby, more focus on the AIE effect is necessary. 

Next, the dual emissive pyrene and tetraphenylethylene-containing amphiphilic [2]rotaxanes were reported with a aggregation-induced static excimer emission (AISEE) property and applied in human lung fibroblasts MRC-5 cellular imaging studies [93]. However, this work did not provide in depth discussions on the bioimaging-based therapeutic application. Likewise, Lo and co-workers reported self-assembly of the poly-L-Lysine-based nanoparticle containing cross-linked pyrenes (PLL) in a pH-triggered drug release study [94]. 

The PLL was used as a nano-carrier to encapsulate Doxorubincin (a chemotherapeutic drug) and to form co-assembled nanoparticles (**PLLD**). Co-assembly in the **PLLD** nanoparticles can be explained in terms of *π–π* stacking, hydrophobic and H-bonding interactions as shown in Figure 15. During pH-triggered drug release (at pH 6.5 and 4–5), the **PLLD** nanoparticles effectively suppressed the tumor in both the in vivo and in vitro interrogations. However, this paper did not provide any information about the AIE effect on the in vivo and in vitro imaging, which requires further attention. The optical properties and imaging applications of the important pyrene conjugates discussed in this section are summarized in Table 2.

## 4. AIE-Tuned Bioimaging from Pyrene-Based Sensory Probes

The pyrene-based derivatives (see Figure 16a–f) with specific analyte sensing ability and AIE-based bioimaging is illustrated in this section. For instance, Srinivasan et al. demonstrated utilization of a pyrene-based Schiff base molecule **C27** (see Figure 16a; noted as “**PS**” in the original reports) towards sensing applications of Fe^3+^ and picric acid and in the AIE-active bioimaging studies [95,96]. Due to its biocompatibility and the AIE effect, incubation of the proposed pyrene conjugate **C27** in diverse cellular lines, such as Hela, A549 (male lung cancer cells) and MCF-7, results in blue emissive cell lines as seen in Figure 17A.

Moreover, the **C27** also displayed strong binding with the bovine serum albumin (BSA) protein, which assured feasibility applications in theranostics. The results from both reports suggest exceptional utility of the **C27** in bioimaging; however, more investigations are required to justify its applicability in theranostics. More recently, a pyrene-based Schiff base **C28** (see Figure 16b; noted as “**5-DP**” in the original report) was demonstrated in the phenolic-nitroaromatic explosives detection and AIE mediated cellular imaging studies [97]. 

The **C28** in a 1:9 THF-water ratio tends to form a *H*-type nano-aggregation as illustrated in Figure 17B. Due to *π–π* stacking, fluorescence of the **C28** is enhanced with a peak shift from 407 to 469 nm. To apply the **C28** in bioimaging more effectively, it was composited with the P123 co-polymers to control the nanoaggregates size (from 484 to <200 nm) and to reduce toxicity and enhance biocompatibility as well. After incubation of the **C28**-P123 composite in Hela cell lines, bright blue emissive image is observed as seen in Figure 17C. This is a good design with multiple applications; however, attention on the therapeutic application is essential in future research.

The one-pot synthesized pyrene-based Schiff base derivatives **C29**–**C32** (see Figure 16c–e; **FBP**, **PCS1**, **PCS2** and **PT2** are the individual representations in their original reports) were reported in metal ions sensing and AIE-based cellular imaging studies [98,99,100]. Among them, the **C29** (detect Fe^3+^, Al^3+^ and Cr^3+^ ions) displayed a blue emissive cell lines in the absence of metal ions due to the AIE effect [98]. However, this report lacked discussions on the AIE-based bioimaging. Both the **C30** and **C31** formed a *J*-type nanoaggregation at 80%/60% *f_w_* with increased particle sizes of 151.7 ± 19.1 and 252.9 ± 63.6 nm from their original sizes of 11.4 ± 1.2 and 24.2 ± 4.2 nm at 0% *f_w_*, respectively, by means of the AIE effect [99]. 

After 12 h incubation of the **C30** and **C31** in Raw264.7 cells, bright blue emissive cells are visualized as shown in Figure 18A. In fact, the presence of the free thiol in the **C30** displayed greater imaging ability than that of the **C31** with the disulfide unit. Regarding the low toxicity and biocompatibility, the **C30** and **C31** can be both used in bioimaging studies and in Fe^3+^, Al^3+^ and Cr^3+^ ions detection and imaging. However, more data are required to justify bioimaging and therapeutic features of the **C30** and **C31**.

Recently, a pyrene-conjugate Schiff base **C32** (see Figure 16e) was reported for Zn^2+^ and tyrosine detection in a solution state and an organic thin film transistor (OTFT) [100]. The probe **C32** is low toxicity and displays the bioimaging ability to the Zn^2+^ and tyrosine. Note that the **C32** in acetonitrile (CH_3_CN) also possesses the AIE effect, which shows red-shifted (from 452 to 468 nm) emission enhancement with increasing *f_w_* from 0 to 97.5% and forms nanofibers (via *J*-type nanoaggregation) via nanoparticles assembly. 

To ensure the bioimaging ability of AIE state, the **C32** is incubated in B16−F10 cell lines (skin cancer cell line) from 1 to 6 h to observe blue emissive cells as illustrated in Figure 18B. Moreover, the probe **C32** delivered a blue emissive zebra fish image by means of the AIE effect, which indicated its bioimaging ability in in vitro and in vivo studies. This is an impressive design with multiple directions, such as the AIE-based bioimaging, sensors in solution, cellular imaging and OTFT device. However, attention on the AIE-tuned bioimaging and therapeutics is still required.

Two pyrene-based probes, **C33** and **C34** (see Figure 16f; **1-PBA** and **2-PBA** are the corresponding representations in their original reports) were reported as effective candidates for the mitochondria imaging studies [101]. This work elaborated a carrier role of the peptide vectors, which carried the **C33** and **C34** molecules into cell organelles. Although this work seems to be impressive, it focused majorly on the free radical and ROS generation but not on the AIE effect, thereby requiring more attention. Optical properties and imaging applications of the important pyrene-based sensory probes discussed in this section are listed in Table 3.

Apart from the demonstrated AIE-based bioimaging utilities, many studies of the pyrene derivatives focused only on the analyte tuned cellular imaging but not the AIE effect [102,103,104,105,106]. On the other hand, some studies of the pyrene-based moieties have described their AIE properties towards diverse optoelectronic applications [107,108,109,110,111] and their potential as the bioimaging candidates in the future.

## 5. Design Requirements, Advantages and Limitations

### 5.1. Design Requirements

To design and develop the pyrene-based molecules as the effective AIE-tuned bioimaging/theranostics candidates, following points must be considered:(1)The molecules must be hydrophobic in nature to be able to induce the AIE in diverse water fractions. Attachment of more hydrophilic units like peptides and cationic salts generation can lead to nanostructures formation; however, careful optimization is required to avoid loss of the AIE features.(2)Since the molecules are designed for the AIE-facilitated bioimaging studies, they should possess low toxicity and are viable in biological environment. If the molecules are designed for the long-term tracking purpose, biocompatibility of the molecules must be justified by the 3-(4,5-dimethylthiazol-2-yl)-2,5-diphenyltetra-zolium bromide (MTT) assay for extended time intervals.(3)Introducing the peptides and polypeptides may enhance biocompatibility of the designed moiety; however, its stability in cellular environment must be attested before subjecting to any imaging investigation. Multiple cleavage of the peptide/polypeptide linked pyrene molecules may lead to loss of imaging or affect the cell culture environment, which results cell death.(4)For efficient energy transfer from the donor pyrene moiety to the acceptor dye in FRET and AIE-tuned two-photon imaging studies, selection of the suitable biocompatible acceptor is necessary. Similarly, for the composited pyrene-moiety/acceptor dye-based FRET/AIE system, optimization of the dye ratio is mandatory in determining the suitable composition for applicability in bioimaging studies.(5)For reactive AIE-based cancerous bio-analytes (such as GSH, Cys and Hcy) detection in cells, the pyrene-based probes must be designed with essential binding units or cleaving units (in FRET mechanism) to initiate the AIE in biological environment.(6)Lastly, if the pyrene moiety is designed for the theranostics drug delivery and bioimaging studies, the molecule should have the higher drug loading ability for conveying drug into specified tumor environment and possess the AIE property for imaging the tumor suppression.

### 5.2. Advantages

The pyrene-based AIE-active molecular design has some advantages over the bioimaging and theranostics as listed below:(1)Due to existence of the fused aromatic structure, *π–π* stacking and self-assembling nature, probability of AIEE occurring by the pyrene containing molecules is high. Under certain circumstance they tend to form emissive nanostructures, which are comparable with recently developed nanomaterials, such as nanocluster, quantum dots, MOFs, etc. [112,113,114,115,116].(2)By attaching biocompatible units, such as peptides, polypeptide and inorganic nanostructures, toxicity of the AIE-active pyrene derivatives can be reduced to be comparable to other therapeutic nanomaterials, such as conjugated polymers, MOFs, carbon-dots, nanoclusters, etc. [117,118,119,120,121,122].(3)Since cell culture is majorly conducted in aqueous environment, the pyrene-based molecules can easily tune their AIE properties at certain water fractions, thereby is effective in bioimaging studies.(4)By means of the AIE mechanism, the pyrene-based small molecules can detect theranostics biothiols, such as GSH, Cys and HCy, as demonstrated in intracellular imaging in in vivo and in vitro studies. Similarly, the DNA conjugated pyrene scaffold was found to be effective towards the cancerous miRNA detection.(5)In the FRET and AIE-based bioimaging studies, the pyrene containing molecules acted as efficient donors in the energy transfer process due to strong *π*-electronic clouds, which can be noted as an additional advantage over other aromatic systems.(6)By conjugating the pyrene moieties with low toxic inorganic/polymeric nanostructures, effective drug delivery and tumor suppression were witnessed via intracellular imaging by means of the AIE of pyrene derivatives.

### 5.3. Limitations

The AIE-active pyrene-conjugated probes for bioimaging and theranostics applications also have certain limitations as described below:(1)Precipitation may form at higher water fractions due to the hydrophobic nature of the pyrene-based AIE-active materials, which will affect the intracellular environment and long-term tracking studies.(2)Covalent linking of excess hydrophilic units with the pyrene derivatives may affect the AIE properties and imaging ability.(3)Self-aggregation/self-assembly of the pyrene containing cationic dye may also lead to aggregation-induced quenching (ACQ), which limits the design towards bioimaging studies.(4)In hybrid nano-drug delivery systems, high concentrated loading of the pyrene derivatives over the proposed nanomaterials may lead to loss of biocompatibility.(5)The NIR dye and pyrene composites proposed in the FRET and AIE-based bioimaging and theranostics studies are limited by the composition ratios, toxicity and biocompatibility.(6)Therapeutic applicability of the pyrene-conjugates is also limited by the pathological and physiological conditions, which need careful optimization.

## 6. Conclusions and Perspectives

In this review, various pyrene-conjugated probes and nanosystems towards bioimaging and theranostics were comprehensively described in three separate sections as noted below. First, applications of the bis-pyrene-conjugated nanosystems towards the AIE-tuned pH-sensitive endocytosis process, bioimaging, PTT, PDT and therapeutics were broadly covered. Moreover, the roles of the bis-pyrene nanoaggregates in the bioimaging and theranostics studies were discussed in detail. Secondly, utilities of many pyrene-based derivatives and dyes towards in vivo and in vitro imaging applications were clarified with their merits and flaws in the AIE-based discussions. The pyrene-conjugated DNA towards cancerous miRNA detection via the AIE effect is described for the readers. 

Thirdly, future directions in AIE-tuned bioimaging studies of the AIE-active pyrene-based small molecules with certain analyte-sensing properties were clearly depicted. Feasible design requirements, advantages and limitations of the pyrene-based molecules were provided to help readers to develop innovative pyrene-conjugated systems that are applicable in biological studies. Although the pyrene-conjugated systems are effective in bioimaging and theranostics applications, still there are some important points to be addressed as noted below.
(1)The majority of reports on the pyrene-conjugated systems applied in bioimaging and theranostics studies were described based on the hydrophobic and self-assembly process rather than the AIE effect. For example, the *J*-type nanoaggregates were formed from the bis-pyrene derivatives to induce emission enhancement in intracellular studies. This phenomenon was generally explained based on the self-assembly but not the AIE effect. Future investigations should rectify this misinterpretation.(2)To authenticate the bis-pyrene-conjugated probes as potential candidates in the AIE-tuned state-of-the-art pH-responsive endocytosis process, attention and supportive reports are required.(3)Applying the bis-pyrene-conjugated cyanine dye in photoacoustic studies appears to be a major research trend, and thus continuous research must be conducted toward the direction of developing commercialized biomedical devices.(4)FRET and AIE-tuned two-photon imaging and therapeutic studies from the bis-pyrene derivatives still need in-depth investigations in the following fields in the future, including biocompatibility, tumor suppression and mice-based theranostics investigations.(5)Reports on the metal ions regulated AIE effect on the pyrene-conjugated system towards the cancerous cell imaging and therapeutics are insufficient, thereby, requiring more attention.(6)Few reports on applying the cancerous analyte- (such as biothiols) induced AIE of pyrene derivatives in bioimaging studies are available, which should be the focus for researchers.(7)Thus far, only one report can be found in applying the pyrene-conjugated system to detect intracellular autophagy via the AIE mechanism. This subject requires greater attention.(8)The pyrene-based derivatives and their hybrid systems towards the AIE-tuned photo thermal and photo dynamic therapies still lack strong evidence and mechanistic explanations, which requires great efforts to rectify.(9)Fewer reports are available to justify the AIE effect of the pyrene-DNA conjugate applied in the cancerous miRNA detection. Concerning its impact on cancer cell detection, similar innovative designs must be continuously pursued.(10)In-depth discussions are missing regarding the intracellular imaging applications using the AIE-active pyrene-based small molecular probes and rotaxanes, which should be addressed with clarity in the future.(11)More experiments are mandatory to justify the AIE effect of the pyrene derivatives and their cationic dye salts towards antibacterial imaging, bioimaging and therapeutic utilities.(12)Many pyrene-containing probes with stimuli/analyte responsive features have not been investigated on the AIE effect, which should be clarified for the research community.

To summarize, the AIE-active pyrene conjugates appear to be promising candidates in bioimaging and theranostics studies. Currently, many research groups are studying and developing novel pyrene-containing designs. Some of the results may lead to major breakthroughs in the development of commercial biomedical devices in the future.

## Figures and Tables

**Figure 1 biosensors-12-00550-f001:**
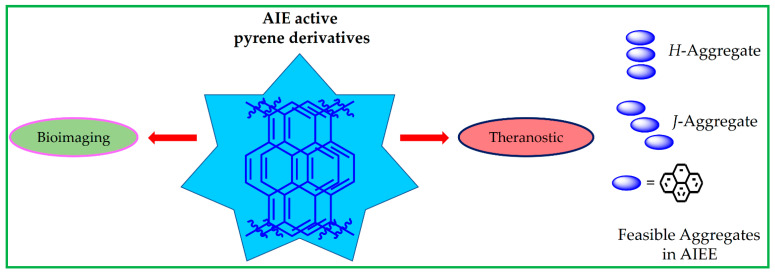
Schematic of AIE-active pyrene derivatives towards bioimaging and theranostics applications and representations of feasible *H*- or *J*-aggregates.

**Figure 2 biosensors-12-00550-f002:**
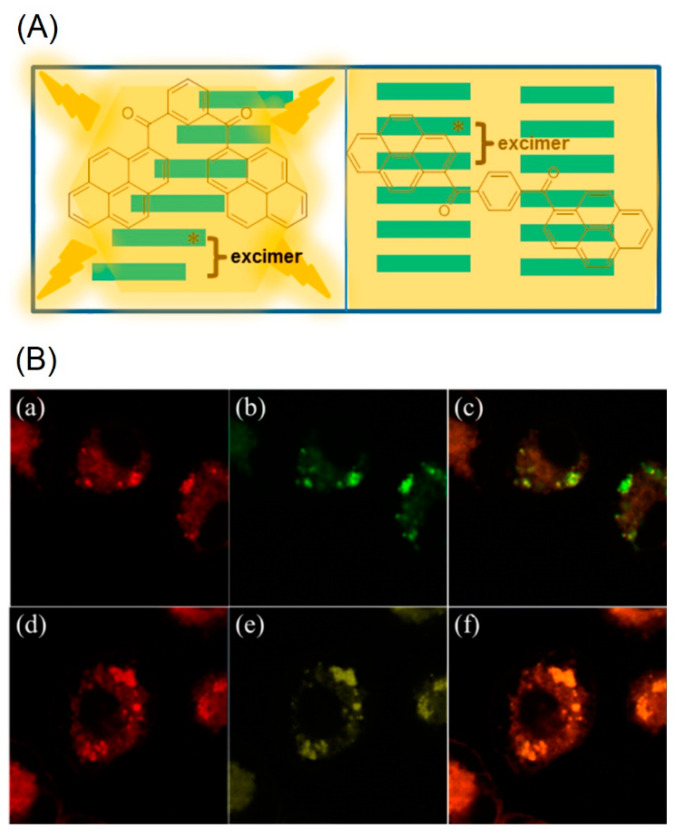
(**A**) Schematic representation of excimer formation by **BP1** and **BP2.** (**B**) Colocalization of LysoTracker red and nanoaggregates. KB cells are double labeled with LysoTracker red (**a**,**d**) and nanoaggregates ((**b**) stands for **BP1** and (**e**) stands for **BP2**). The overlayed images are presented in (**c**) (overlayed image of (**a**,**b**)) and (**f**) (overlayed image of (**d**,**e**)). Excitation wavelength: 577 nm (for LysoTracker red) and 350 nm (for **BP1** and **BP2**) (Reproduced with the permission from Ref. [65]).

**Figure 3 biosensors-12-00550-f003:**
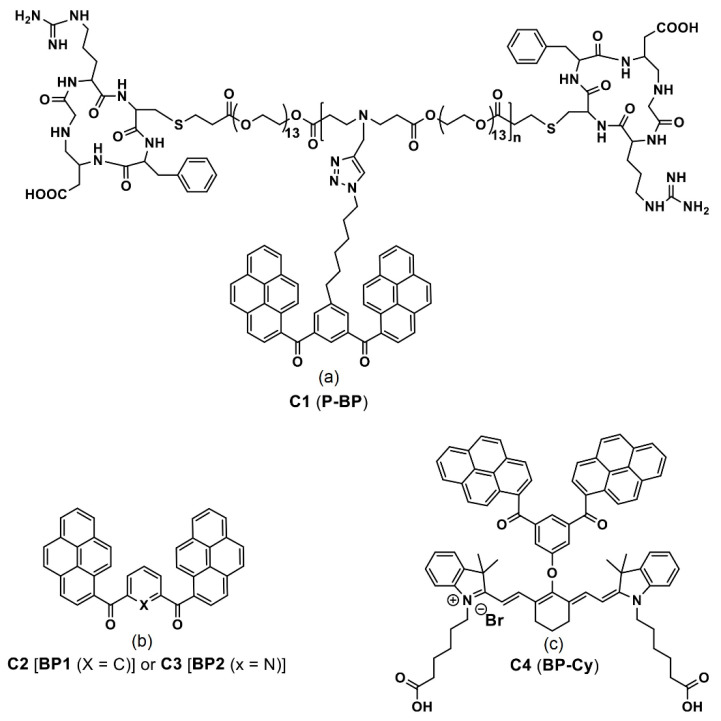
(**a**–**c**) **C1**–**C4** are the chemdraw structures of bis-pyrene-conjugated polymers, small molecules and dyes molecules (Note: the original representation or numbering of the molecules were provided in the brackets along with respective reference numbers).

**Figure 4 biosensors-12-00550-f004:**
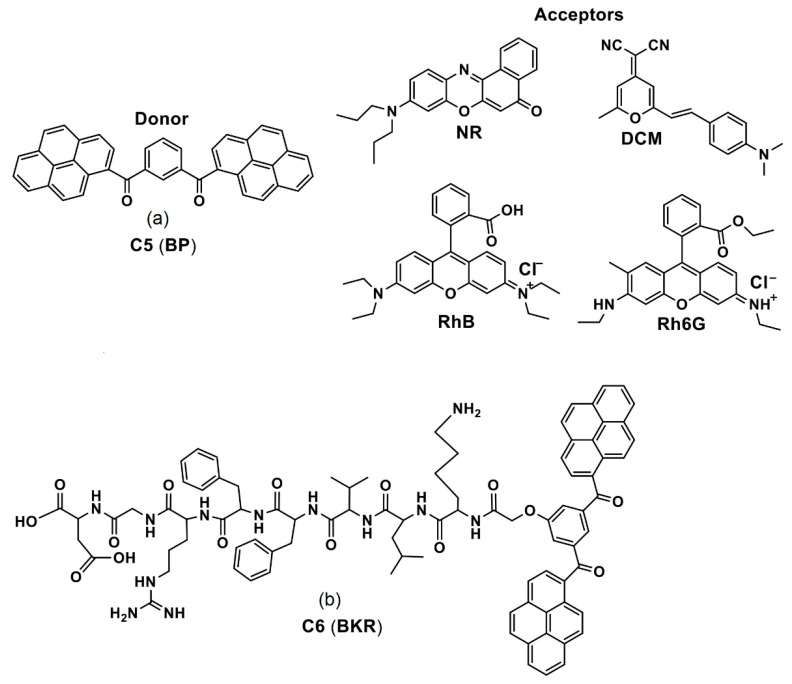
(**a**,**b**) **C5** and **C6** are the chemdraw structures of bis-pyrene donor (along with acceptors used) and bis-pyrene-conjugated peptide, respectively. (Note: the original representation or numbering of the molecules were provided in the brackets along with respective reference numbers).

**Figure 5 biosensors-12-00550-f005:**
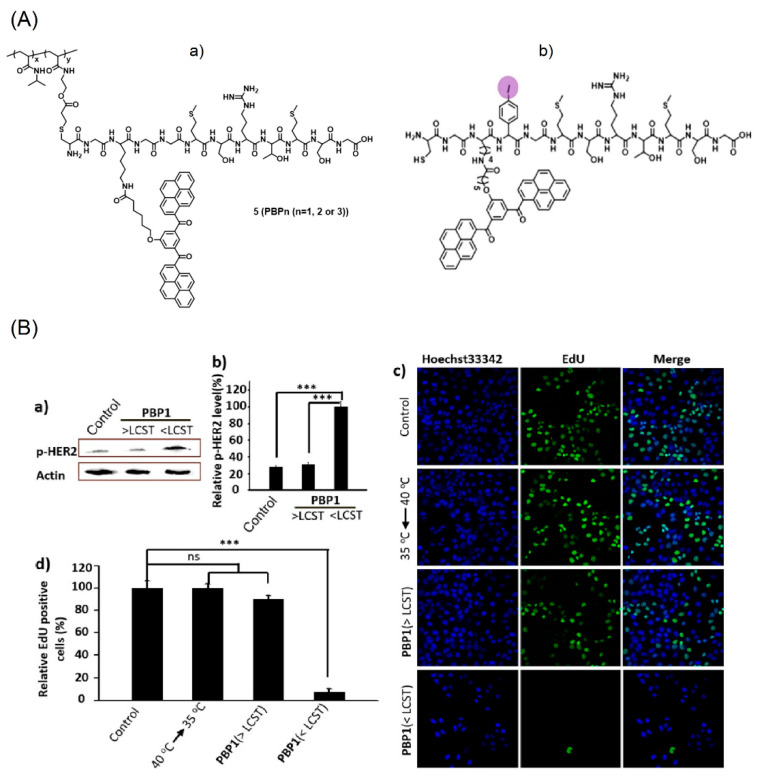
(**A**) (**a**,**b**) structures of HPB containing polymers. (**B**) (**a**) Western blot analysis of HER2 phosphorylation induced by HBP aggregation at 40 °C (>LCST) and 35 °C (<LCST). (**b**) Bar chart represents the normalized band intensity ratio of p-HER2 as quantified by ImageJ. Actin was used as a loading control. (**c**) Detection of EdU incorporated into the DNA of cultured SK-BR-3 cells after different treatment by fluorescence microscopy. (**d**) Histogram shows significant decrease of EdU-labeled cells in the **PBP1** treated SK-BR-3 cells at 35 °C (<LCST), and temperature change alone had no effect on cells. EdU-labeled cell number of different treated groups were compared. *** *p* < 0.001; ns, not significant; one-way ANOVA for indicated comparison. The results are the mean values ± SD of independent experiments performed in triplicate (Reproduced with the permission from Ref. [72]).

**Figure 6 biosensors-12-00550-f006:**
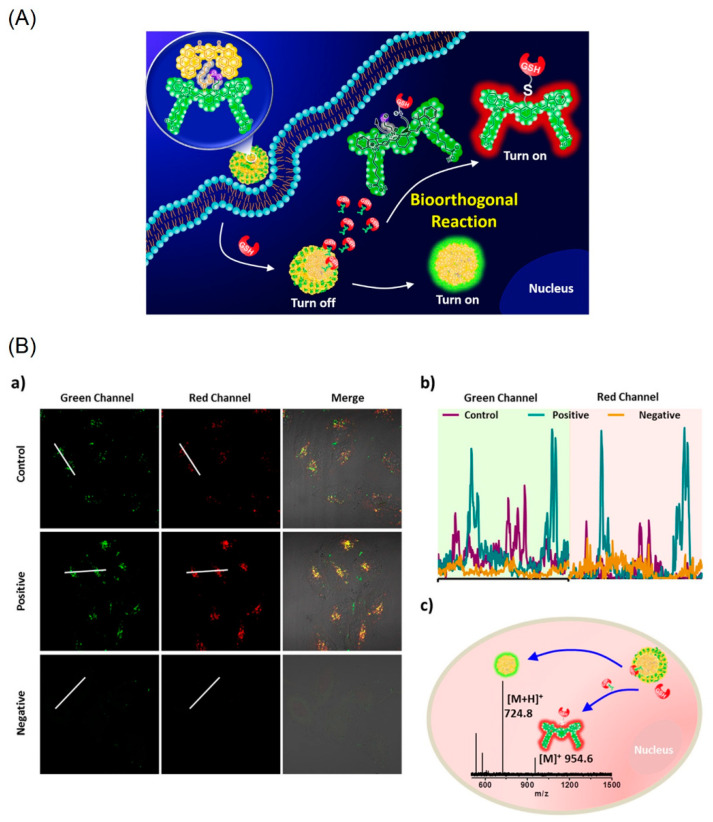
(**A**) Schematic illustration of a new GSH-bio-orthogonal reaction that enable to decipher the nanoemitters of cyanine-pyrene dye (**1**). Resulting in turn-on binary fluorescence signals with a large shift excitation/emission profiles in living systems. (**B**) GSH-based bio-orthogonal reaction for intracellular application. (**a**) CLSM images of the nanoemitters 1 (30 μM) in MCF-7 cells (control), LPA (a lipoic acid, a GSH enhancer) pretreated cells (1.8 mM, 24 h, positive) and NEM (N-ethylmalemide a GSH scavenger) treated cells (500 μM, 2 h, negative). (**b**) Representative line plot of MCF-7 cells and the corresponding fluorescence signal distribution based on the white line in images. (**c**) The MALDI-TOF spectrum of MCF-7 cell lysates after incubation with the nanoemitters 1 (50 μM) for 3 h (Reproduced with the permission from Ref. [73]).

**Figure 7 biosensors-12-00550-f007:**
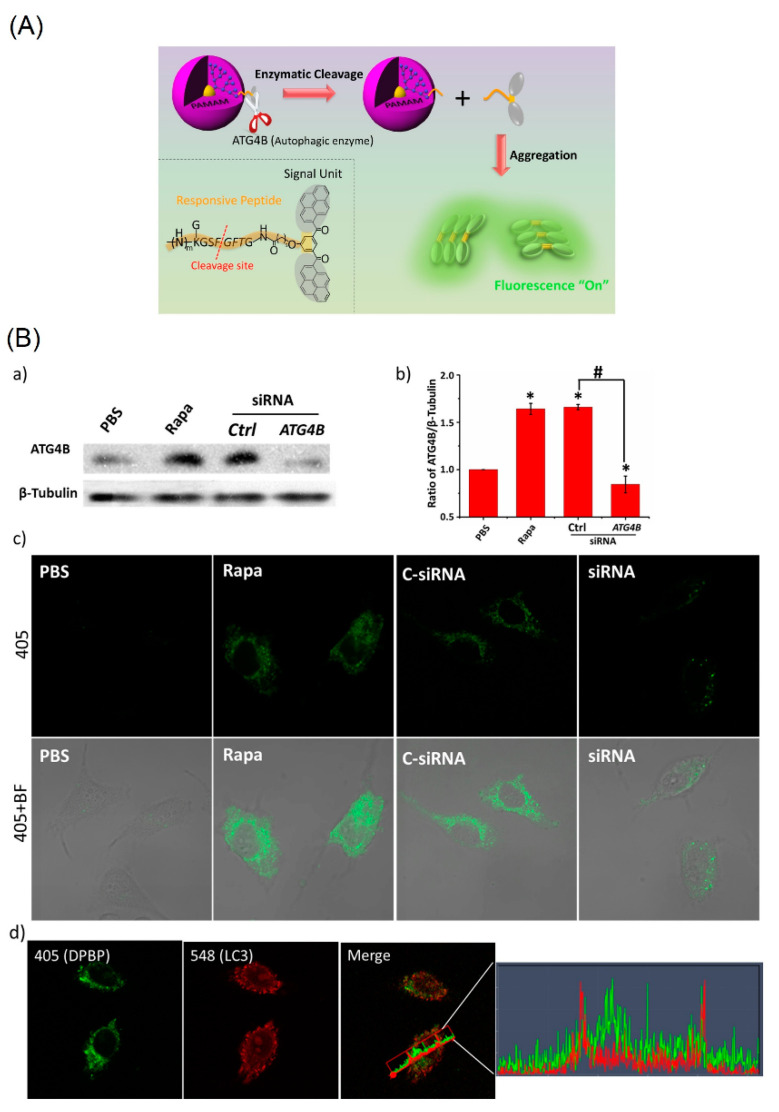
(**A**) Schematic illustration of **DPBP** as a bioprobe for autophagy detection. (**B**) Analyzing the specificity of **DPBP** to ATG4B in living cells. (**a**) Western blot of ATG4B expression in MCF-7 cells. (**b**) Quantitative analysis of band intensity of ATG4B from (**a**). (**c**) ATG4B activity was detected by **DPBP** in MCF-7 cells. MCF-7 cells were treated with 10 μL of PBS in a regular culture medium (lane 1); 10 μL Rapa (1.0 μM) (lane 2); transfection of a control siRNA (lane 3); and transfection of a siRNA against ATG4B (lane 4), followed by **DPBP** (100 μg/mL) for 2 h. (**d**) Correlation plot of **DPBP** and LC3. An overlap coefficient of 0.46 (0: no colocalization and 1: all pixels colocalized) confirmed that the BP spots from **DPBP** did not colocalize with LC3. Statistical significance: * *p* < 0.05 and # *p* < 0.05, one-way ANOVA for indicated comparison. 405: λ_ex_ = 405 nm, λ_em_ = 525 ± 50 nm; BF: bright field. (Reproduced with the permission from Ref. [74]).

**Figure 8 biosensors-12-00550-f008:**
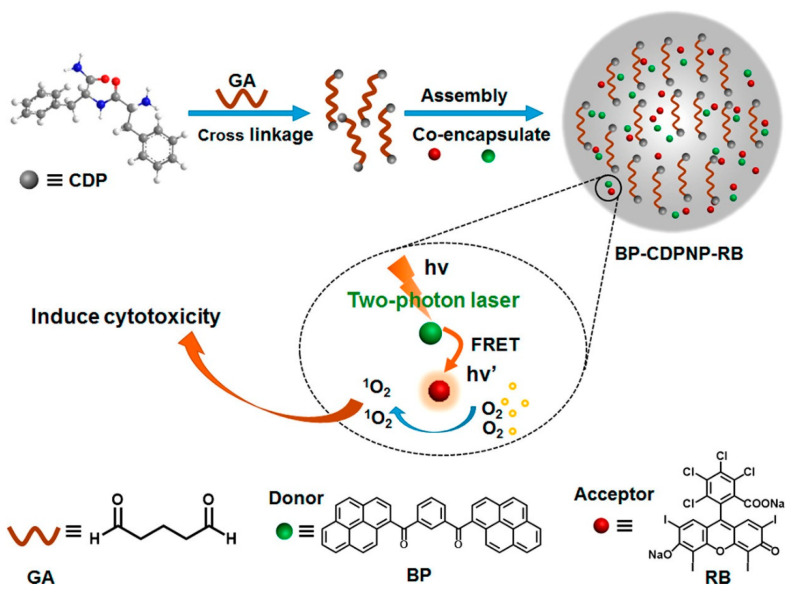
Schematic Illustration of fabrication of **BP-CDPNP-RB** nanosystem and application of TPA-PDT via intraparticle FRET (Reproduced with the permission from Ref. [75].

**Figure 9 biosensors-12-00550-f009:**
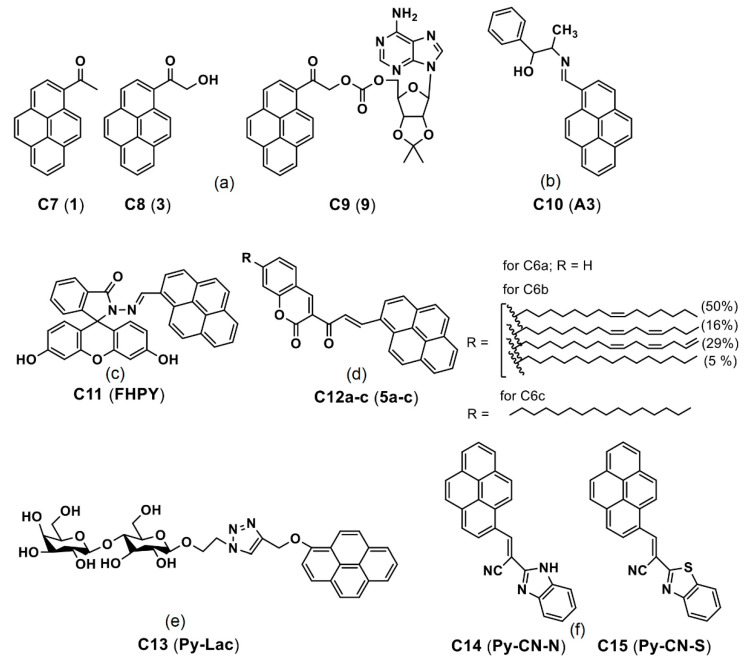
(**a**–**f**) **C7**–**C15** are the chemdraw structures of pyrene-based AIEgens, small molecules and dyes reported for *in vitro*/*in vivo* imaging (Note: the original representation or numbering of the molecules were provided in the brackets along with respective reference numbers).

**Figure 10 biosensors-12-00550-f010:**
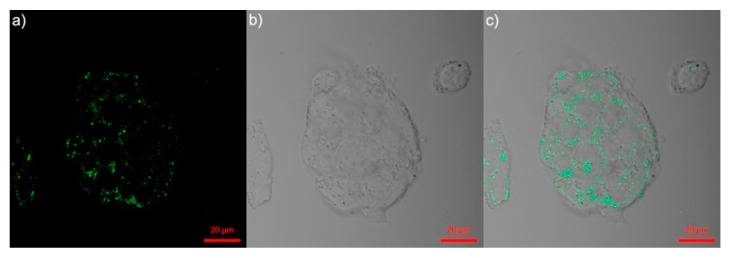
Confocal microscopic images of Hep G2 cells after incubation with **C13** (noted as “**Py-Lac**” in the original report) (10 μM) for 2 h ((**a**) excited at 780 nm, (**b**) bright field and (**c**) merged image of (**a**,**b**)) (reproduced with the permission from Ref. [82]).

**Figure 11 biosensors-12-00550-f011:**
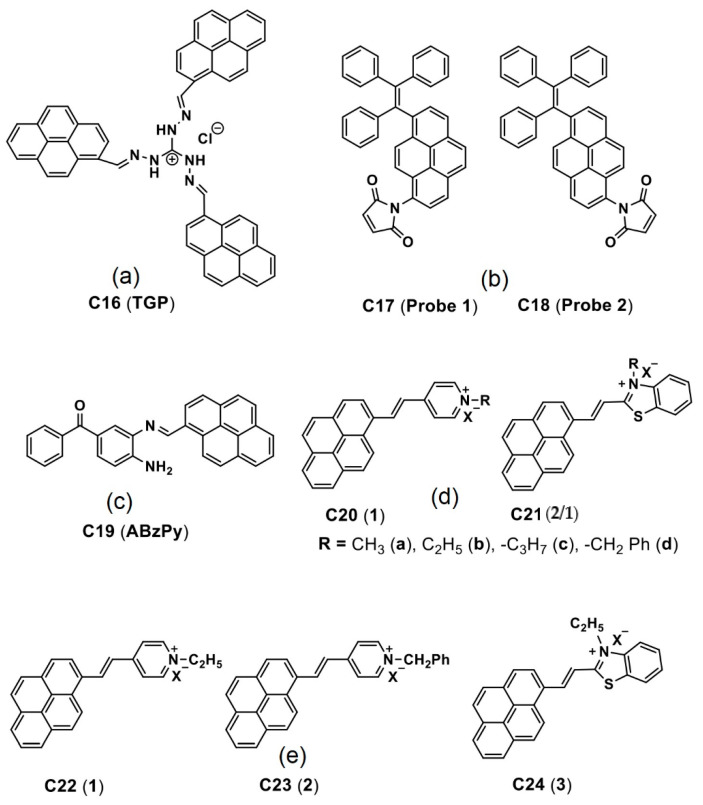
(**a**–**e**) **C16**–**C24** are the chemdraw structures of pyrene-based AIEgens, small molecules and dyes reported for *in vitro*/*in vivo* imaging (Note: the original representation or numbering of the molecules were provided in the brackets along with respective reference numbers).

**Figure 12 biosensors-12-00550-f012:**
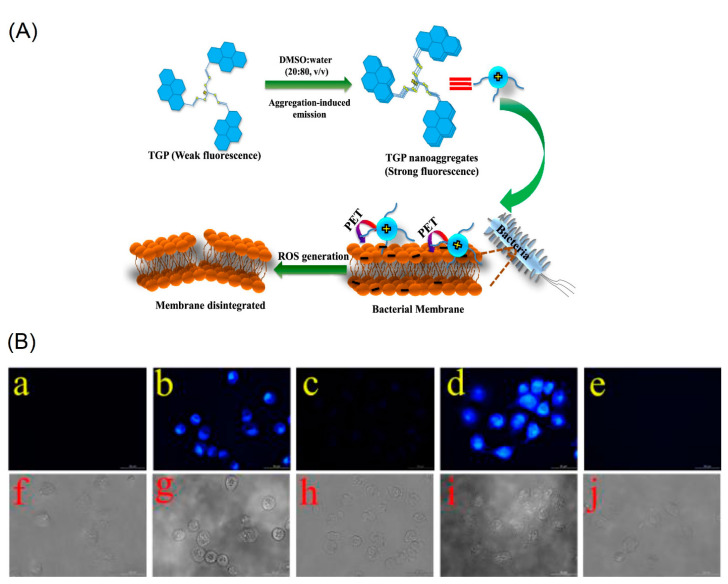
(**A**) Schematic representation of **C16** (noted as “**TGP**” in the original report) in the molecular state, their self-assemblies in the aggregated state and proposed interactions with bacterial membrane exhibiting broad-spectrum antimicrobial activity (Reproduced with the permission from Ref. [84]). (**B**) Fluorescence images of H1299 cells. (**a**) Fluorescence images of cell. (**b**) Fluorescence images of cells incubated with probe 1 (10 μM) for 20 min at 37 °C. (**c**) Fluorescence images of cells pretreated with N-methylmaleimide (100 μM) for 1 h at 37 °C and then incubated with **C17** (5 μM; noted as “**Probe 1**” in the original report) for 20 min at 37 °C. (**d**) Fluorescence images of cells incubated with probe 2 (5 μM) for 20 min at 37 °C. (**e**) Fluorescence images of cells pretreated with N-methylmaleimide (100 μM) for 1 h at 37 °C and then incubated with **C18** (1 μM; noted as “**Probe 2**” in the original report) for 20 min at 37 °C. (**f**–**j**) are the corresponding bright field images of (**a**–**e**) (Reproduced with the permission from Ref. [85]).

**Figure 13 biosensors-12-00550-f013:**
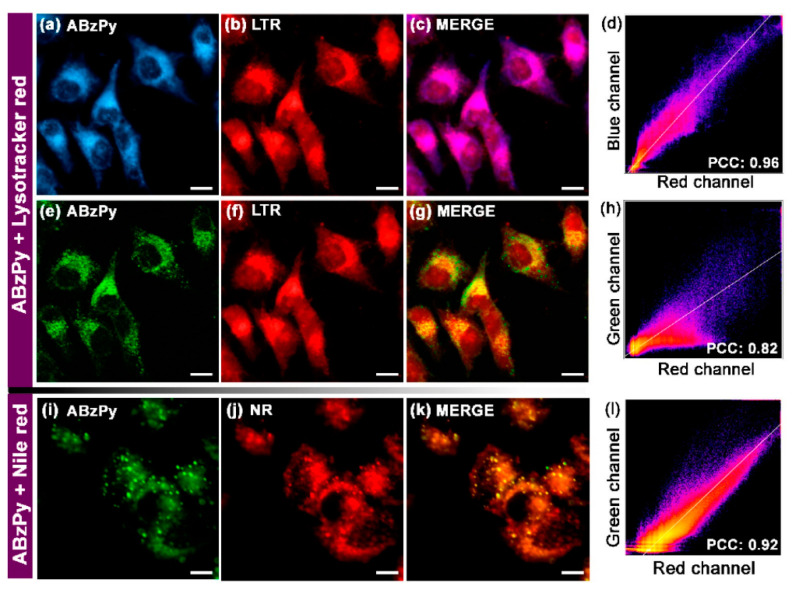
Fluorescence microscopy images of HeLa cells (**a**–**c**,**e**–**g**) incubated with the aqueous dispersion of **C19** (represented as “**ABzPy**” in the original report) and lysotracker red (LTR) for 30 min: (**a**) DAPI filter (blue channel, λ_ex_ = 335–383 nm, λ_em_ = 420–470 nm; **C19**), (**b**) DsRed filter (red channel, λ_ex_ = 538–562 nm, λ_em_ = 570–640 nm; LTR), (**c**) merged image of panels (**a**,**b**) and (**d**) the colocalization plot of red channel (LTR) vs. blue channel (**C19**) with Pearson’s coefficient of colocalization (PCC) value of 0.96 depicting specific targeting of lysosomes by **C19**. (**e**) GFP filter (green channel, λ_ex_ = 450–490 nm, λ_em_ = 500–550 nm; **C19**), (**f**) DsRed filter (red channel, λ_ex_ = 538–562 nm, λ_em_ = 570–640 nm; LTR), (**g**) merged image of panels (**e**,**f**) and (**h**) the colocalization plot of red channel (LTR) vs. green channel (**C19**) with PCC value of 0.82 depicting the reduced level of colocalization. Fluorescence microscopy images of HeLa cells (**i**–**k**) incubated with the aqueous dispersion of **C19** and Nile red (NR, a lipid droplet tracker dye) for 30 min: (**i**) GFP filter (green channel, λ_ex_ = 450–490 nm, λem = 500–550 nm; **C19**), (**j**) DsRed filter (red channel, λ_ex_ = 538–562 nm, λ_em_ = 570–640 nm; NR), (**k**) merged image of panels (**i**,**j**) and (**l**) the colocalization plot of red channel (NR) vs. green channel (**C19**) with PCC value of 0.92 depicting specific targeting ability of **C19** to lipid droplets. Scale = 10 μm (Reproduced with the permission from Ref. [86]).

**Figure 14 biosensors-12-00550-f014:**
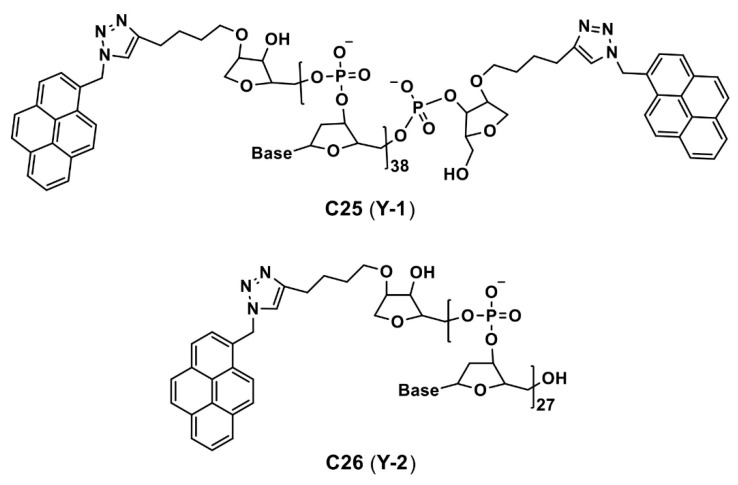
**C25** and **C26** are the structures of Y-shaped pyrene containing DNA nanoprobe reported for microRNA imaging (Note: the original representation or numbering of the molecules were provided in the brackets along with respective reference number).

**Figure 15 biosensors-12-00550-f015:**
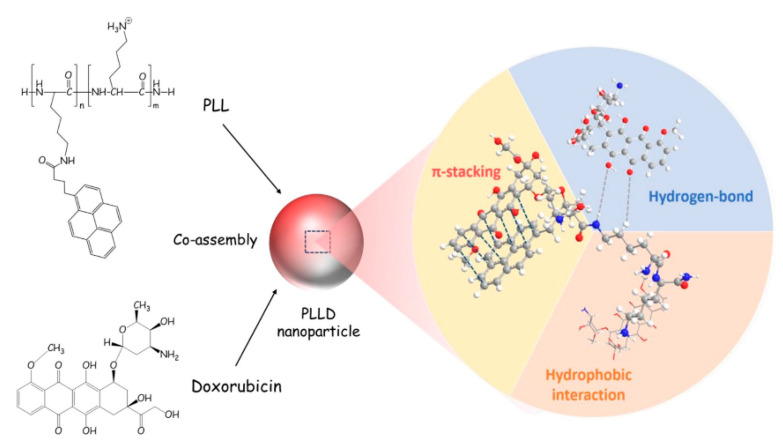
Schematic figure of cooperative self-assembly of PLL and therapeutics (Reproduced with the permission from Ref. [94]).

**Figure 16 biosensors-12-00550-f016:**
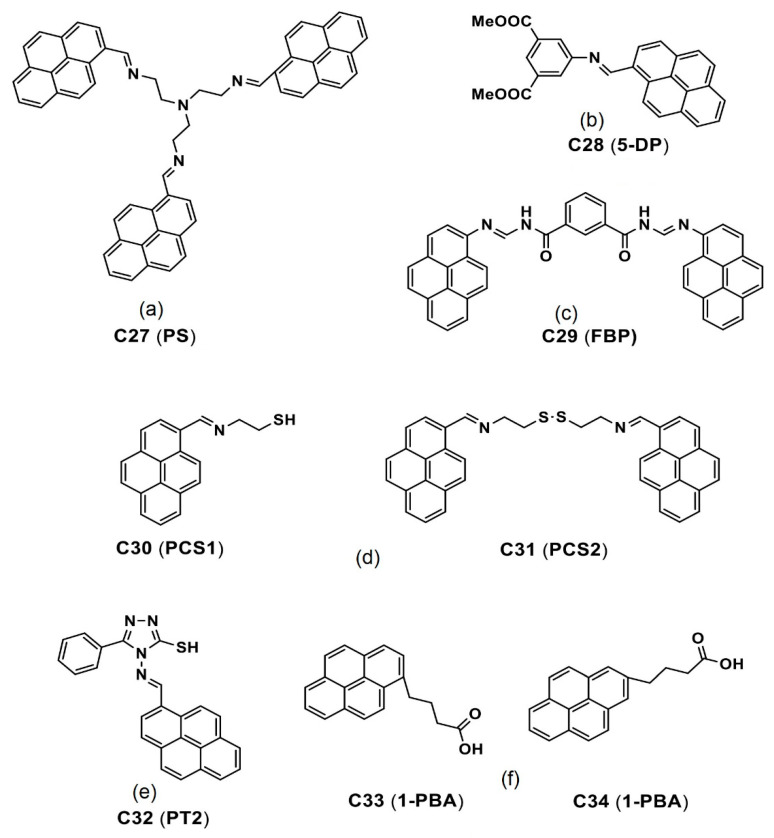
(**a**–**f**) **C27**–**C34** are the chemdraw structures of pyrene-conjugated small molecules (Note: the original representation or numbering of the molecules were provided in the brackets along with respective reference numbers).

**Figure 17 biosensors-12-00550-f017:**
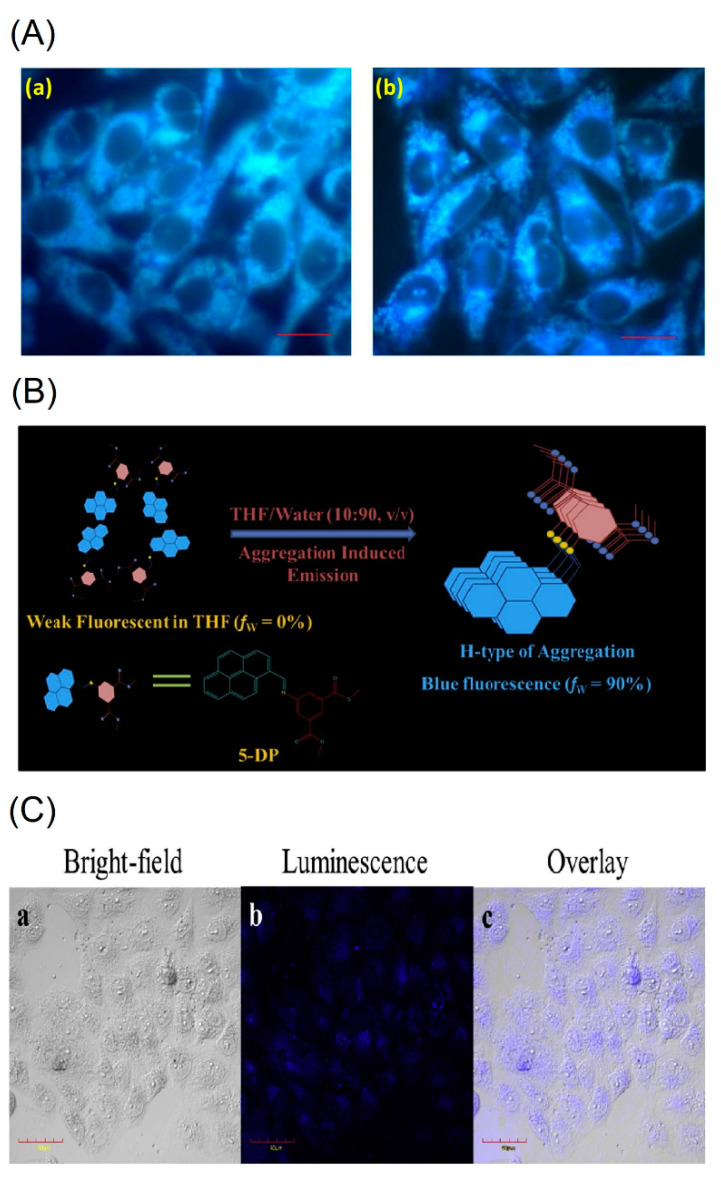
(**A**) Fluorescent microscopic observation of (**a**) A549 & (**b**) MCF-7 cells treated with the **C27** (represented as “**PS**” in the original report) aggregates. Scale bar represents 20 μm (Reproduced with the permission from Ref. [95]). (**B**) Schematic representation of the formation of nanoaggregates of **C28** (represented as “**5-DP**” in the original report) in THF-Water aqueous mixture through *H*-Type of aggregation. (**C**) Confocal imaging of HeLa cells incubated with 40 μg mL^−1^ of **C28**-P123 nanoaggregates for 4 h at 37 °C, scale bar 40 μm (**B**,**C** are reproduced with the permission from Ref. [97]).

**Figure 18 biosensors-12-00550-f018:**
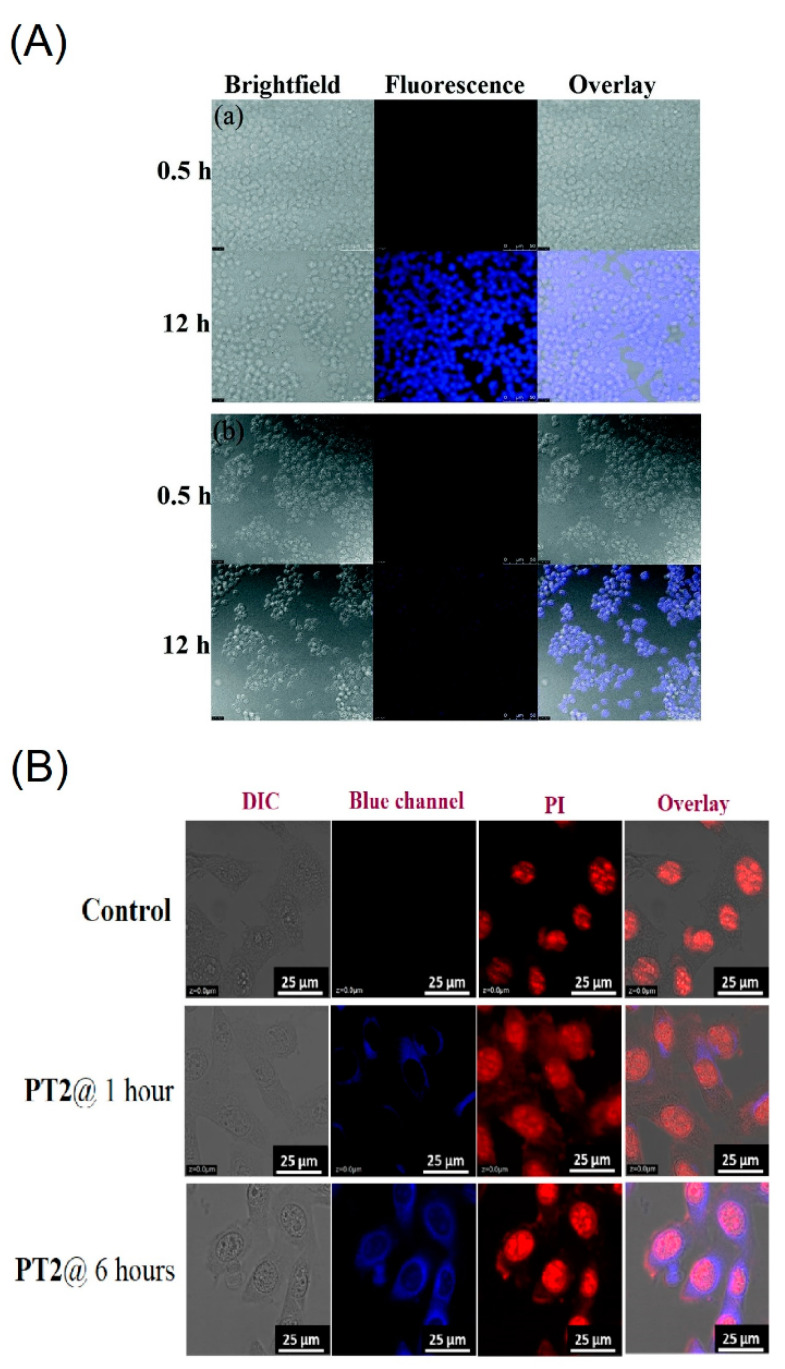
(**A**) Fluorescence images of Raw264.7 cells treated with (**a**) **C30** and (**b**) **C31** (assigned as “**PCS1** and **PCS2**” in the original report) at 0.5 and 12 h, respectively. Bright field image (**left**); fluorescence image (**middle**); merged image (**right**). The scale bar is 50 mM (Reproduced with the author’s permission of Ref. [99]). (**B**) Cellular images of **C32** (represented as “**PT2**” in the original manuscript) at 1 and 6 h, respectively; Cell line: B16-F10; Scale bar: 25 µm (Reproduced with the permission of Ref. [100]).

**Table 1 biosensors-12-00550-t001:** Photophysical properties and imaging applications of important bis-pyrene compounds/systems.

Compound/System	λ_abs_ (nm)	λ_em_ (nm)	λ_em_ Stokes Shift (nm)	Φ_em_ (%)	Applications	Ref
**BP1** and **BP2**	342 and 378 ^a^ 418 (for both) ^b^	506 and 394 ^a^ 512 and 542 ^b^	n/a 6 and 48 ^b^	1.1 (for both) ^a^ 32.6 and 10.5 ^b^	Lysosome imaging	[65]
**C1** (**P-BP**)	NA NA	537 ^a^ 418 ^b^	n/a 119 ^b^	NA NA	Lysosome endocytic pH imaging	[66]
**C2** and **C3** (**BP1** and **BP2**) with polymer	342 and 378 ^a^ NA	506 and 394 ^a^ 512 and 533 ^b^	n/a 6 and 39 ^b^	1.1 (for both) ^a^ NA	Lysosome endocytic pH imaging	[67]
**C4** (**BP-Cy**)	790 ^a^ 850	812 ^a^ >825 ^b^	n/a >13 ^b^	24.6 NA	In vivo PA imaging	[68]
**C6** (**BKR**)	395 ^a^ 411 ^b^	519 ^a^ 528 ^b^	n/a 9 ^b^	NA NA	Cancer cell imaging	[71]
**DPBP**	NA	525 ^a^ 525 ^b^	n/a NA	NA NA	Intracellular autophagy imaging	[74]

NA, n/a = Not available, not applicable; ^a^ Solution state; ^b^ Aggregated state; Brackets-denotation in original reports.

**Table 2 biosensors-12-00550-t002:** Photophysical properties and imaging applications of important pyrene conjugates.

Compound	λ_abs_ (nm)	λ_em_ (nm)	λ_em_ Stokes Shift (nm)	Φ_em_ (%)	Applications	Ref
**C10** (**A3**)	385 ^a^ 394 ^b^	404 ^a^ 505 ^b^	n/a 101 ^b^	NA NA	Cancer cell imaging	[79]
**C11** (**FHPY**)	371 ^a^ 391 ^b^	451 ^a^ 470 ^b^	n/a 19 ^b^	12 97	Cancer cell imaging	[80]
**C16** (**TGP**)	365 ^a^ 419 ^b^	413 ^a^ 467 ^b^	n/a 54 ^b^	11.7 36	Bacterial imaging	[84]
**C19** (**ABzPy**)	365 ^a^ 420 ^b^	450 ^a^ 575 ^b^	n/a 125 ^b^	<1 >4	Cancer cell imaging	[86]
**C25** and **C26** (**Y-1** and **Y-2**)	260 (for both) ^a^ 260 (for both) ^b^	400 ^a^ (for both) ^a^ 480 (for both) ^b^	n/a 80 ^b^	NA NA	Cancer cell imaging	[90]

NA, n/a = Not available, not applicable; ^a^ Solution state; ^b^ Aggregated state; Brackets-denotation in original reports.

**Table 3 biosensors-12-00550-t003:** Photophysical properties and imaging applications of important pyrene-based sensory probes.

Compound	λ_abs_ (nm)	λ_em_ (nm)	λ_em_ Stokes Shift (nm)	Φ_em_ (%)	Applications	Ref
**C27** (**PS**)	350–370 ^a^ 390–420 ^b^	430–435 ^a^ 475 ^b^	n/a >35 ^b^	0.7/0.1 ^a^ 48/54 ^b^	Cancer cell imaging	[95,96]
**C28** (**5-DP**)	378 ^a^ 365 ^b^	407 ^a^ 469 ^b^	n/a 62 ^b^	1 ^a^ 67 ^b^	Cancer cell imaging	[97]
**C30** (**PCS1**) and **C31** (**PCS2**)	356 and 352 ^a^ 364 and 357 ^b^	421 and 425 ^a^ 465 and 469 ^b^	n/a 44 (for both) ^b^	1.1 and 1.5 ^a^ 55 and 85	Cancer cell imaging	[98]
**C32** (**PT2**)	401 ^a^ 413 ^b^	453 ^a^ 468 ^b^	n/a 15	1 ^a^ 68 ^b^	Cancer cell imaging	[100]

NA, n/a = Not available, not applicable; ^a^ Solution state; ^b^ Aggregated state; Brackets-denotation in original reports.

## Data Availability

Not applicable.

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
