# Peer review of "Pyrene-Based AIE Active Materials for Bioimaging and Theranostics Applications"

_biosensors, 2022, doi:10.3390/bios12070550_

Round 1

Reviewer 1 Report

Sun and co-workers comprehensively summarized a review of pyrene-conjugated probes and nanosystems for bioimaging and theragnostic applications. Here, the authors mainly focused on three different aspects: first, applications of the bis-pyrene-conjugated nanosystems for the AIE-tuned pH-sensitive endocytic process, bioimaging, PTT, PDT, and therapeutics. Second, the utility of many pyrene-based derivatives and dyes for in vivo and in vitro imaging applications. Third, the future directions in the AIE tuned bioimaging studies of the AIE-active pyrene-based small molecules with certain analyte sensing properties have been clearly manifested. Furthermore, the authors have done a good job of documenting the feasible design requirements, advantages, and limitations of the pyrene-based molecules to help the readers develop innovative pyrene-conjugated systems applicable in biological studies. Overall, the review presented here will play a significant role in the future design and development of AIE-based pyrene conjugates for the vast majority of biological applications.

I would recommend the publication of this interesting review in the journal of Biosensors after considering the minor revisions.   

In revision, the following comments and corrections should be considered.

1.     The authors should tabulate the photophysical properties (absorption maximum, emission maximum, stokes shifts, fluorescent quantum yields, etc.,) of some important pyrene chromophores.

2.     In the table, authors should report the photophysical properties before (in solution state) and after (aggregation state), in this way readers can take some advantage for their own applications.

3.     In section 3, AIE active pyrene conjugates for in-vivo/in-vitro imaging” authors claimed that in vivo imaging of the pyrene chromophore, it would be better to include some in vivo imaging data where pyrene has been used extensively.

4.     In Figure 17B, please change the background color from black to something else to make it clearer.

5.     Please clean up (bond angles and bond lengths) the ChemDraw structures of Figures 4, 9, 11, 14, and 16 to make it clear. 

.

Author Response

Reviewer 1

Sun and co-workers comprehensively summarized a review of pyrene-conjugated probes and nanosystems for bioimaging and theragnostic applications. Here, the authors mainly focused on three different aspects: first, applications of the bis-pyrene-conjugated nanosystems for the AIE-tuned pH-sensitive endocytic process, bioimaging, PTT, PDT, and therapeutics. Second, the utility of many pyrene-based derivatives and dyes for in vivo and in vitro imaging applications. Third, the future directions in the AIE tuned bioimaging studies of the AIE-active pyrene-based small molecules with certain analyte sensing properties have been clearly manifested. Furthermore, the authors have done a good job of documenting the feasible design requirements, advantages, and limitations of the pyrene-based molecules to help the readers develop innovative pyrene-conjugated systems applicable in biological studies. Overall, the review presented here will play a significant role in the future design and development of AIE-based pyrene conjugates for the vast majority of biological applications. I would recommend the publication of this interesting review in the journal of Biosensors after considering the minor revisions.   

We are very grateful to the reviewer for providing the opportunity to improve the standard of this review article. We have followed the reviewer’s suggestion and rectified the pointed issues.

 In revision, the following comments and corrections should be considered.

  1. The authors should tabulate the photophysical properties (absorption maximum, emission maximum, stokes shifts, fluorescent quantum yields, etc.,) of some important pyrene chromophores.

“Author Response”

As suggested by the reviewer, Tables 1-3 are delivered in sections 2-4.

  1. In the table, authors should report the photophysical properties before (in solution state) and after (aggregation state), in this way readers can take some advantage for their own applications.

“Author Response”

As recommended by the reviewer, the photophysical properties of few important pyrene compounds in solution and aggregation state are provided in Tables 1-3.

  1. In section 3, AIE active pyrene conjugates for in-vivo/in-vitro imaging” authors claimed that in vivo imaging of the pyrene chromophore, it would be better to include some in vivo imaging data where pyrene has been used extensively.

“Author Response”

We agree with the reviewer’s comment, but it will take a while to get permissions for providing additional in vivo imaging data/figures. Nevertheless, we have modified the title of section 3 as “AIE active pyrene conjugates for bioimaging”.

  1. In Figure 17B, please change the background color from black to something else to make it clearer.

“Author Response”

Figure 17B was taken from Ref [97] and not drawn/modified by us. Thus, changing the background color is not possible. However, we do try our best to further improve the figure resolution.

  1. Please clean up (bond angles and bond lengths) the ChemDraw structures of Figures 4, 9, 11, 14, and 16 to make it clear. 

“Author Response”

For Chemdraw structures in Figures 3, 4, 9, 11, 14 and 16, the bond angles and bond lengths were present in the original reports. None of the figures were redrawn by ourselves. But, in the revision the clarity and resolution of those chemdraw figures are further improved.

Reviewer 2 Report

The manuscript presented by Shellaiah  and Sun focused on applications of pyrene-based AIE materials in bioimaging as well as therapeutics and diagnostics. First, Authors give general information about  the  AIE compounds and describe properties of pyrene derivatives. Then, Authors systematically describe the chosen examples: bis-pyrene derivatives, pyrene conjugates, pyrene-based sensory probes. The next chapter contains clues regarding designing effective AIE pyrene-based compounds as well as advantages and limitations of such systems. The manuscript closes the list of conclusions and perspectives. At the end, there is of course the list of 104 references. 

As mentioned, Authors focused on pyrene-based AIE materials. The pyrene belongs to π-conjugated molecules and such kind of molecules exhibit the different emissive behaviors: ACQ: aggregation-caused quenching, AIE: aggregation-induced emission, RTP: room-temperature phosphorescence. Especially pyrene is known to form fluorescent excimers and I think Authors should add some information about pyrene excimers - at least mention about it in introduction. In main text, Authors also give some information about J or H aggregates, for unfamiliar readers would be also good to add some information about this phenomena prior describing the chosen examples.

I have doubts to the organizations of the chapters: the first chapter is about bis-pyrene derivatives and it is ok, but the title of 2nd and 3rd chapters are not good in terms of their content. Besides reading the manuscript I actually not sure which of AIE pyrene systems are realy used for theranostic applications  and which exhibit only potential applications. The Table summaryzing the information about described systems  (i.e. kind of derivatives, biomaging in cells, therapeutic or diagnostic applications, reference) would be very helpful for reader. When it comes to choosing examples/ citations I generally have no comments, however the Authors could cite manuscript entitled "Recent Advances in AIEgens for Metal Ion Biosensing and Bioimaging" by Li et al. in Molecules 2019. By the way I recommend Authors to add the new examples of AIE pyrene systems  (published after 2019) for metal ion biosensing to their manuscript.

Author Response

Reviewer 2:

The manuscript presented by Shellaiah and Sun focused on applications of pyrene-based AIE materials in bioimaging as well as therapeutics and diagnostics. First, Authors give general information about the AIE compounds and describe properties of pyrene derivatives. Then, Authors systematically describe the chosen examples: bis-pyrene derivatives, pyrene conjugates, pyrene-based sensory probes. The next chapter contains clues regarding designing effective AIE pyrene-based compounds as well as advantages and limitations of such systems. The manuscript closes the list of conclusions and perspectives. At the end, there is of course the list of 104 references. 

We thank the reviewer for giving valuable comments, which allow us to improve our review article. We have followed the reviewer’s suggestion and rectified the pointed issues.

As mentioned, Authors focused on pyrene-based AIE materials. The pyrene belongs to π-conjugated molecules and such kind of molecules exhibit the different emissive behaviors: ACQ: aggregation-caused quenching, AIE: aggregation-induced emission, RTP: room-temperature phosphorescence. Especially pyrene is known to form fluorescent excimers and I think Authors should add some information about pyrene excimers - at least mention about it in introduction. In main text, Authors also give some information about J or H aggregates, for unfamiliar readers would be also good to add some information about these phenomena prior describing the chosen examples.

“Author Response”

As recommended by the reviewer, information regarding J or H aggregates has been added in the introduction section and Figure 1.

I have doubts to the organizations of the chapters: the first chapter is about bis-pyrene derivatives and it is ok, but the title of 2nd and 3rd chapters are not good in terms of their content. Besides reading the manuscript I actually not sure which of AIE pyrene systems are realy used for theranostic applications and which exhibit only potential applications. The Table summaryzing the information about described systems (i.e. kind of derivatives, biomaging in cells, therapeutic or diagnostic applications, reference) would be very helpful for reader. When it comes to choosing examples/ citations I generally have no comments, however the Authors could cite manuscript entitled "Recent Advances in AIEgens for Metal Ion Biosensing and Bioimaging" by Li et al. in Molecules 2019. By the way I recommend Authors to add the new examples of AIE pyrene systems (published after 2019) for metal ion biosensing to their manuscript.

“Author Response”

Based on the reviewer’s comment, we have modified the titles of the 2nd and 3rdchapters as “AIE active pyrene conjugates for bioimaging” and “AIE tuned bioimaging from pyrene-based sensory probes”. Tables 1-3 have been revised to provide photophysical properties of a few important pyrene compounds in solution and aggregation state (see sections 2-4). The paper (Recent Advances in AIEgens for Metal Ion Biosensing and Bioimaging" by Li et al. in Molecules 2019) by Li et al., is cited as Ref [25]. In fact, examples in section 4 demonstrates the AIE based bioimaging from analyte sensing probes. Adding the AIE pyrene systems for metal ion biosensing may not fit within this article’s scopes. Many pyrene systems towards metal ion biosensing were not from based on the metal ions tuned AIE effect. Moreover, separate reviews to that direction were readily available. Thus, it is not essential to include those examples.